# Assessment of Physical Activity Using Waist-Worn Accelerometers in Hospitalized Heart Failure Patients and Its Relationship with Kansas City Cardiomyopathy Questionnaire

**DOI:** 10.3390/jcm10184103

**Published:** 2021-09-11

**Authors:** Yasuyuki Shiraishi, Nozomi Niimi, Ayumi Goda, Makoto Takei, Takehiro Kimura, Takashi Kohno, Masataka Kawana, Keiichi Fukuda, Shun Kohsaka

**Affiliations:** 1Department of Cardiology, Keio University School of Medicine, Tokyo 160-8582, Japan; figarofuga@gmail.com (N.N.); kimura@z7.keio.jp (T.K.); kfukuda@a2.keio.jp (K.F.); sk@keio.jp (S.K.); 2Department of Cardiovascular Medicine, Kyorin University Faculty of Medicine, Tokyo 181-8611, Japan; ayumix34@yahoo.co.jp (A.G.); kohno.ta@gmail.com (T.K.); 3Department of Cardiology, Tokyo Saiseikai Central Hospital, Tokyo 108-0073, Japan; makoto_tk@hotmail.com; 4Department of Medicine, Division of Cardiovascular Medicine, Stanford University, Stanford, CA 94305, USA; mkawana@stanford.edu

**Keywords:** heart failure, physical activity, Kansas City Cardiomyopathy Questionnaire, accelerometer, cardiopulmonary exercise testing

## Abstract

The health benefits of physical activity have been widely recognized, yet there is limited information on associations between accelerometer-related parameters and established patient-reported health status. This study investigated the association between the waist-worn accelerometer measurements, cardiopulmonary exercise testing (CPX), and results of the Kansas City Cardiomyopathy Questionnaire (KCCQ) in heart failure (HF) patients hospitalized for acute decompensation. A total of 31 patients were enrolled and wore a validated three-axis accelerometer for 2 weeks and completed the short version of the KCCQ after removing the device. Daily step counts, exercise time (metabolic equivalents × hours), and %sedentary time (sedentary time/device-equipped time) were measured. Among the measured parameters, the best correlation was observed between %sedentary time and the KCCQ overall and clinical summary scores (r = −0.65 and −0.65, each *p* < 0.001). All of the individual domains of the KCCQ (physical limitation, symptom frequency, and quality of life), with the exception of the social limitation domain, showed moderate correlations with %sedentary time. Finally, oxygen consumption assessed by CPX demonstrated only weak associations with the accelerometer-measured parameters. An accelerometer could complement the KCCQ results in accurately assessing the physical activity in HF patients immediately after hospitalization, albeit its correlation with CPX was at most moderate.

## 1. Introduction

The health benefits of physical activity have been widely recognized to minimize disease incidence and progression, such as heart failure (HF), and further to optimize patients’ health status [1,2,3]. Self-reported questionnaires are frequently used to assess physical activity and quality of life (QoL), with minimal time investment, costs, and participant burden, which favors their use in both epidemiological studies and large-scale clinical trials. However, the disadvantage of these questionnaires is that they are prone to recall bias, response shift, and social desirability bias and therefore, they are likely to over or underreport physical activity levels. Accelerometers have advanced epidemiologic research on physical activity by providing objective and continuous measurement of physical activity in free-living conditions [4]. Waist-worn accelerometers have become especially popular due to low participant burden.

Previously, secondary analysis of three randomized controlled trials targeting patients with HF with preserved ejection fraction reported that lower QoL scores were associated with impaired physical activity levels assessed by an accelerometer (e.g., average daily accelerometry units or accelerometry hours active per day) [5]. In addition, a previous small-scale pilot study (*n* = 10) described that device-measured daily step counts had moderate correlations with QoL scores and conventional clinical parameters, including 6-min walk distance, in patients hospitalized for newly diagnosed HF with reduced ejection fraction [6]. However, there is still limited information on clinical factors associated with physical activity/QoL and for which accelerometer-measured parameters are useful indicators associated with exercise capacity and QoL scores in HF patients [7].

Accurate measurement of physical activity in HF patients is necessary to assess its effects on various health outcomes and to estimate the effectiveness of the medical treatments and interventions [1]. In addition, identifying physically inactive patients can help with implementing an effective exercise intervention program [2]. Within this context, we aimed to (1) investigate clinical factors associated with physical activity measured by waist-worn accelerometers and QoL and (2) examine the overall association between the level of physical activity assessed by the self-reported questionnaire, the accelerometer, and cardiopulmonary exercise testing (CPX) in a group of HF patients recently discharged after acute decompensation regardless of left ventricular ejection fraction.

## 2. Materials and Methods

### 2.1. Study Cohort and Sample

Between January 2019 and October 2020, we analyzed data from 34 hospitalized HF patients who required hospitalization for acute decompensation and were successfully discharged to their residences in three tertiary care hospitals within the metropolitan area of Tokyo, Japan. This study was carried out as a pilot phase feasibility study in the West Tokyo Heart Failure Registry (WET-HF) [8,9]. Briefly, the WET-HF was launched in January 2006 and added several institutions to several facilities (8 tertiary hospitals) in April 2018. Individual cardiologists made the clinical diagnosis of HF at each institution based on the Framingham criteria [10] and the level of natriuretic peptides (B-type natriuretic peptide [BNP] > 100 pg/mL) at the time of hospitalization [11]. Additional inclusion criteria for the present study were repeated hospitalization for HF and/or increased doses in the ambulatory setting within a year, no clinically overt dementia (expected to have reasonable adherence to wearing the accelerometer), and the New York Heart Failure (NYHA) functional class II or III. Patients who had a life expectancy of not more than a year due to other diseases, such as malignancy, were excluded from the present study. Furthermore, patients with physical/mental disability or locomotive syndromes were also excluded.

Each site’s institutional review board approved the study protocol (Keio University Institutional Review Boards [permission number: 20170292]), and the research was conducted in accordance with the principles of the Declaration of Helsinki. Written informed consent was obtained from each subject before the study began. The study was planned and performed in accordance with the Strengthening the Reporting of Observational Studies in Epidemiology guidelines.

### 2.2. Clinical Data Acquisition

Data on demographics, medical history, clinical frailty, laboratory and other tests (such as electrocardiogram and echocardiogram), medications, procedures, and clinical outcomes during hospitalization and after discharge were recorded (Figure 1). BNP levels were measured at the time of both hospitalization and discharge. The left ventricular ejection fraction was calculated using the modified Simpson’s method. Ultrasound cardiography was performed by highly experienced cardiologists or clinical technologists during the index hospitalization. The NYHA functional class was evaluated at the time of discharge by the treating cardiologists at each institution. The Clinical Frailty Scale (CFS) is a semiquantitative tool that provides a generally accepted clinical definition of frailty [12]. The CFS score was calculated by face-to-face assessment with patients/families and cardiologists during the index hospitalization.

### 2.3. Accelerometry

Daily physical activity was measured using a waist-worn triaxial accelerometer (Active Style Pro HJA-750C, Omron Co., Ltd., Kyoto, Japan) [13,14]. Anteroposterior, mediolateral, and vertical acceleration measurements were obtained during each physical activity at a rate of 32 Hz to 12-bit accuracy. Each of the three signals from the triaxial accelerometer was passed through a high-pass filter with a cut-off frequency of 0.7 Hz to remove the gravitational acceleration component. Activity data were calculated with the integral of the accelerometer output’s absolute value for each of the three axes using acceleration signals over a 10 s time interval. The ratios of the unfiltered to filtered total acceleration and filtered vertical to horizontal acceleration were calculated to determine the cut-off value for the classification of locomotive activities and non-locomotive activities such as household and occupational activities, which resulted in an almost 100% accurate demarcation for the daily 11 different activities [13]. Furthermore, metabolic equivalents (METs) determined by this triaxial accelerometer have been reported to closely correlate with METs calculated using energy expenditure measured by indirect calorimetry [14]. No detection of acceleration signal for longer than 60 consecutive minutes was defined as non-wear time. Data were included in the present analyses if participants had at least 10 h of wear time per day for a minimum of 4 days [15].

Investigators at each site instructed individual patients to continuously wear the accelerometer around the waist for 2 weeks, except when sleeping, bathing, or showering. When physicians judged that the patient’s condition was not yet fully stabilized, they allowed patients to wear an accelerometer after achieving a sufficient stabilization in the ambulatory setting. The timing to wear an accelerometer was left to the physicians’ discretion based on physical examinations and biomarkers including natriuretic peptide levels. The daily activity level was reported as total step counts (generally more than 2 METs), exercise time (defined as METs × hours), and %sedentary time (defined as sedentary time/device-equipped time in the daytime). Sedentary time was defined as physical activity of less than or 1.5 Mets. Physical activity of <3 METs defined as light-intensity physical activity was excluded from the calculation of exercise time [16].

### 2.4. Health-Related Quality of Life

The Kansas City Cardiomyopathy Questionnaire (KCCQ) has a 2-week recall period and includes 23 items that map to seven domains: symptom frequency, symptom burden, symptom stability, physical limitation, social limitation, QoL, and self-efficacy [17]. Furthermore, the short version of the original KCCQ is now available, a 12-item instrument (KCCQ-12). Both versions have been validated across a wide spectrum of HF patients [18]. KCCQ has also received federal certification as a clinical outcome assessment tool, which helps to standardize the patients’ history over time and share more consistent insights on the patients’ well-being with health care systems consisting of multiple medical providers [19,20].

In this study, health status was assessed using the KCCQ-12, which summarized the findings as the overall summary score (OSS) and the clinical summary score (CSS) [18]. Scores ranged from 0 to 100, with higher scores reflecting better health status. Individual parameters in the KCCQ, including physical limitation, symptom frequency, QoL, and social limitation, were also assessed. The physical limitation and symptom frequency domains were merged into CSS that mirrors the key concepts of the NYHA functional class. All domains were combined to obtain the OSS, which is the primary health status outcome in most trials. Patients completed the questionnaire on their own at two time points: (1) after admission and (2) immediately after wearing the accelerometer for 2 weeks. Of the 34 patients, while two could not complete the physical activity assessment due to lost devices, a third patient died and could not answer the KCCQ. Excluding these three patients, the data of 31 were finally analyzed in the present study.

### 2.5. Cardiopulmonary Exercise Testing

All patients underwent CPX after being sufficiently stabilized just before discharge or in the ambulatory setting prior to wearing the accelerometer, except three of them with severe physical frailty. An incremental symptom-limited exercise test was performed with an electromagnetically braked ergometer (Strength Ergo 8, Fukuda Denshi, Tokyo, Japan) according to the ramp protocol. The test consisted of a 2 min resting period, which was followed by 2 min of warm-up at an ergometer setting of 0 W (60 rpm) and testing with a 1 W increase in exercise load every 4–6 s (10–15 W/min), depending on the predicted maximum exercise capacity and in such a way that maximal effort was attained within 8 to 15 min. During the test, heart rate, blood pressure, oxygen saturation, and electrocardiogram were recorded and monitored continuously in all subjects.

During exercise, oxygen consumption (VO_2_), carbon dioxide production (VCO_2_), and minute ventilation (VE) were measured using the 10 s average with a metabolic cart (AE-302S; MINATO, Tokyo, Japan). Peak VO_2_ was calculated as the average VO_2_ during the last 30 s of exercise. The anaerobic threshold (AT) point was determined using the *V*-slope method in addition to the following conventional criteria: VE/VO_2_ increases after registering as flat or decreasing, whereas VE/VCO_2_ remains constant or decreases [21,22]. The VE vs. VCO_2_ slope was calculated from the start of incremental exercise to the respiratory compensation point by least squares linear regression analysis.

### 2.6. Right Heart Catheterization

Right heart catherization (RHC) was performed after achieving clinically sufficient stabilization during the index hospitalization. The decision to perform RHC and its timing was left to individual physicians in charge. Cardiac output was measured by the thermodilution technique if a case of severe tricuspid regurgitation or shunting was evaluated using the estimated oxygen uptake Fick method.

### 2.7. Statistical Analysis

Continuous variables are presented as median (interquartile range (IQR)) and categorical data are presented as absolute numbers (%). For baseline characteristics, the patients were divided into two groups according to the KCCQ-OSS and compared using the Mann–Whitney U test for continuous variables and the chi-square test or Fisher exact test for categorical variables, as appropriate.

To examine the association between the accelerometer-measured parameters, KCCQ, and CPX parameters, the Pearson product-moment correlation coefficient (r) was calculated. We also created a heatmap of the correlation coefficient of accelerometer-measured parameters, KCCQ scores, CPX parameters, BNP, NYHA functional class, and CFS (with ordering of variables based on hierarchical clustering using hclust in R). In addition, we performed a subgroup analysis for patients who received RHC during the index hospitalization. All the tests were two-sided, and values of *p* < 0.05 were considered statistically significant. All statistical analyses were performed using SPSS version 26.0 (SPSS Inc., Chicago, IL, USA) and R (version 4.1.0; R Project for Statistical Computing, Vienna, Austria).

## 3. Results

### 3.1. Patients’ Characteristics

The KCCQ scores at the time of hospitalization and after wearing a waist accelerometer for 2 weeks (at baseline) for the 31 patients with HF were presented in Online Appendix A. The KCCQ scores dramatically improved during the hospitalization (the median length of hospital stay (IQR), 23 days (13–35 days)). The median KCCQ-OSS value at baseline was 60.0 (IQR, 42.7–74.0). Furthermore, the median (IQR) values of the accelerometer-measured daily step counts, exercise time, and %sedentary time were 2604 (926–4247), 2.17 (1.2–3.5), and 76.3% (69.4–81.5%), respectively. All patients had at least 10 h of wear time per day for a minimum of 4 days, and the median (IQR) duration of wearing the accelerometer was 10 (7–14) days.

Table 1 summarizes the baseline characteristics of the patients based on the median KCCQ-OSS. Laboratory findings and medications were assessed at discharge after sufficient inpatient care. The patients were predominantly men (65%), with a median (IQR) age, left ventricular ejectiont fraction, and BNP level of 63 (51–78) years, 32% (24–42%), and 370 (244–437) pg/mL, respectively. Most patients (74%) were classified as having NYHA functional class III at the time of discharge, and the median peak VO_2_ was 11.5 (10.5–15.0) mL/kg/min. Compared with the high KCCQ-OSS group, the patients in the low KCCQ-OSS group had a higher CFS with a lower body mass index (each *p* < 0.05). BNP values decreased considerably in the majority of the patients during the index hospitalization, while the low KCCQ-OSS group had higher levels of blood urea nitrogen and BNP and usage of loop diuretics more frequently. In addition, the low KCCQ-OSS group demonstrated a higher VE vs. VCO_2_ slope value and lower physical activity levels (via accelerometer measurement) compared with the high KCCQ-OSS group.

### 3.2. Association between KCCQ, Accelerometer-Measured Parameters, and CPX Data

Figure 2 shows the correlation of KCCQ scores, accelerometer-measured parameters, CPX parameters, BNP, NYHA class, and CFS. Among accelerometer-measured parameters, the best correlation was seen between KCCQ and %sedentary time: KCCQ-OSS (r = −0.65, *p* < 0.001) and KCCQ-CSS (r = −0.65, *p* < 0.001). Of the KCCQ domains, physical limitation (r = −0.58, *p* < 0.001), symptom frequency (r = −0.56, *p* = 0.001), and QoL (r = −0.49, *p* = 0.005) showed moderate correlations with %sedentary time, while social limitation showed a weak correlation (r = −0.35, *p* = 0.051) (Figure 3). KCCQ showed correlations with similar tendencies with the daily step counts and exercise time.

Figure 4 shows that the peak VO_2_ showed no or weak correlation with each of the accelerometer-measured parameters. Other CPX parameters, including VO_2_ at AT and VE vs. VCO_2_ slope, also indicated no association with the accelerometer-measured parameters. Peak VO_2_ showed modest correlations with the KCCQ results: OSS (r = 0.43, *p* = 0.021) and CSS (r = 0.47, *p* = 0.012). While the peak VO_2_ showed modest correlations with physical limitation (r = 0.41, *p* = 0.03) and symptom frequency domains (r = 0.42, *p* = 0.026), which are merged into CSS, it showed no correlation with the QoL (r = 0.31, *p* = 0.11) and social limitation domain (r = 0.22, *p* = 0.27) (Figure 5). As for VO_2_ at AT and VE vs. VCO_2_ slope, there were no associations with the KCCQ results other than VO_2_ at AT and KCCQ-CSS (r = 0.44, *p* = 0.018), and physical limitation and symptom frequency domains (r = 0.38 and 0.41, *p* = 0.047 and 0.03).

### 3.3. Subgroup Analysis

In 14 patients who were substantially stabilized on RHC (Online Appendix A), a median (IQR) right atrial pressure, pulmonary capillary wedge pressure, and cardiac index were 3 (0–7) mm Hg, 8 (4–22) mm Hg, and 2.4 (2.0–2.9) L/min/m^2^, respectively. In the subgroup analysis for these patients, the associations between accelerometer-measured parameters and KCCQ were directionally similar with the main results (Online Appendix A). Meanwhile, the VO_2_ at peak and AT were also correlated modestly with the KCCQ results (Online Appendix A). In addition, the VO_2_ at peak and AT were significantly correlated with %sedentary time (r = −0.74, *p* = 0.006 for peak VO_2_; r = −0.63, *p* = 0.028 for VO_2_ at AT) but not with daily step counts and exercise time.

## 4. Discussion

Overall, the patients with low scores on the KCCQ questionnaires had more severe congestive status reflected by higher blood urea nitrogen and BNP levels and frequent usage of diuretics. In addition, these patients had a higher grade of frailty. Our analysis demonstrated that accelerometer-measured physical activity in HF patients immediately after their discharge correlated well with health status determined by KCCQ. Of the several accelerometer-measured indicators, %sedentary time showed the best association with KCCQ outcomes. Accelerometer is increasingly recognized as a pertinent portable device directly assessing the daily activity level of the HF patients enrolled in clinical trials.

In patients with HF, lifestyle modification, including physical fitness training, is paramount to improving exercise capacity and QoL [23,24]. The American Heart Association recommends routine assessment and promotion of physical activity and reduction in sedentary time to decrease the risk of cardiovascular diseases and associated healthcare expenditures and improve function and prognosis in HF patients [1,25]. As both health-related QoL and physical activity are important, our findings on correlations between the KCCQ and the objective accelerometer-measured physical activity can be valuable to HF’s clinical practice. Repeated assessments of QoL and physical activity via disease-specific questionnaires are not widely accepted in daily clinical practice due to several challenges (e.g., time, cost, familiarity, and complexity in HF care). Accelerometry can provide real-time and unique information on health status compared to intermittent assessment.

There is an increasing focus on exercise capacity in the management of HF. However, to date, there are no consistent and universally accepted methods and indicators of accelerometer-measured physical activity for direct clinical interpretation. We found a significant correlation between %sedentary time and the KCCQ results. Assessment of time active or inactive per day is easily attainable using commercially available wearable devices with accelerometry. Accelerometry enables the quantitative assessment of sedentary time to reasonably promote and implement the exercise therapy for patients with HF, while KCCQ includes no quantitative assessment of physical activity. On the other hand, our main analysis found no clear association between conventional parameters of exercise capacity (e.g., peak VO_2_ and AT-VO_2_ (submaximal exercise capacity)) and accelerometer-measured physical activity. A disparity is commonly seen between exercise capacity (maximal potential) and actual daily activity [26,27]. The OUTSTEP-HF trial, which compared the effects of sacubitril/valsartan with enalapril on physical activity, demonstrated an inverse relationship between exercise capacity and daily activity after initiating medical interventions [28]. The NEAT-HFpEF trial reported a similar tendency [26]. This may be explained by the lack of motivation and the burden of comorbidities [29]. In addition, there are few studies validating accelerometers in the HF patient population [7]. Therefore, the estimated physical activity derived from healthy adults, such as Mets, may deviate significantly from those obtained in patients with HF [30]. Meanwhile, for patients who were sufficiently stabilized on RHC, %sedentary time strongly correlated with VO_2_ assessed by CPX in the subgroup analysis of our study. The result is consistent with the previous study on stable HF patients in the ambulatory setting [5], and thus, our study indicates the importance of the timing to quantitatively measure physical activity in HF patients. Given that the actual physical activity impacts exercise capacity and other QoL domains, it is important to encourage patients to promote attitudes for the self-management of HF. Activity trackers such as the accelerometer can promote and foster the development of patient-health professional partnerships essential to achieve better quality treatment plans, although there are challenges to their efficient implementation in HF management.

## 5. Limitations

This study has several limitations. First, the patients did not wear the accelerometer throughout the day, and thus, we may have underestimated their daily physical activity. However, we excluded the days when the participants did not wear the accelerometer for more than 10 h (which were few) from the data for analysis. Second is the small sample size, making it difficult to generalize our results to a wider HF patient population. Our patients were more severely ill (repeated hospitalizations, low peak VO_2_ and high VE vs. VCO_2_ slope, and low KCCQ scores) than patients enrolled in the landmark trials on sacubitril/valsartan and dapagliflozin [31,32]. Moreover, it should be considered with caution that no participants with physical/mental disability or locomotive syndrome were included in the present analysis. Third, the patients in the present study were hospitalized for acute decompensation and discharged within a month, and thus, they may excessively restrict physical activity, despite individual education for the self-assessment of HF inclusive of physical activity based on the CPX results. There was also a possibility that the timing of measuring physical activity using an accelerometer may not have been the best (too early). However, the length of hospital stay was long enough that the BNP level decreased sufficiently at the time of discharge in our study. Additionally, there was an interval of approximately 1–2 weeks between the CPX and wearing of the accelerometer, leading to further stabilization of congestive symptoms during the study period. Therefore, this may influence the association with CPX parameters and accelerometer-assessed physical activity.

## 6. Conclusions

Our data suggest that accelerometer-assessed physical activity provides real-time information on health status, which shows a moderate association with the validated HF-specific questionnaire among hospitalized HF patients. The accelerometer complements the KCCQ and is expected to enhance the treatment approach for further improving health-related QoL and physical activity in these patients.

## Figures and Tables

**Figure 1 jcm-10-04103-f001:**
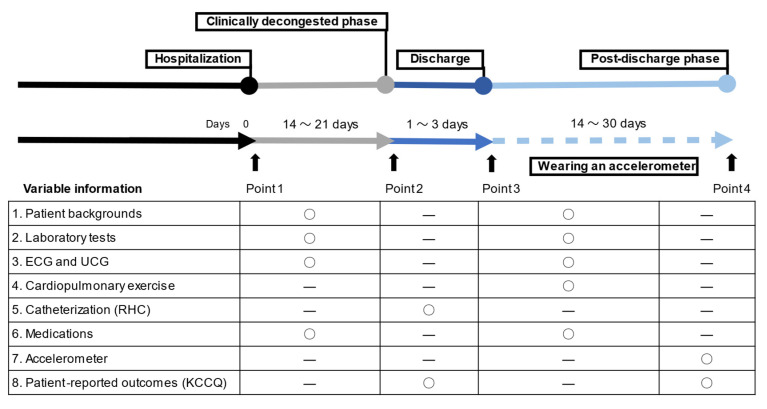
A summary of measured variables. Circles mean the correction point for each item. ECG = electrocardiogram; UCG = ultrasound cardiogram; RHC = right heart catheterization; KCCQ = Kansas City Cardiomyopathy Questionnaire.

**Figure 2 jcm-10-04103-f002:**
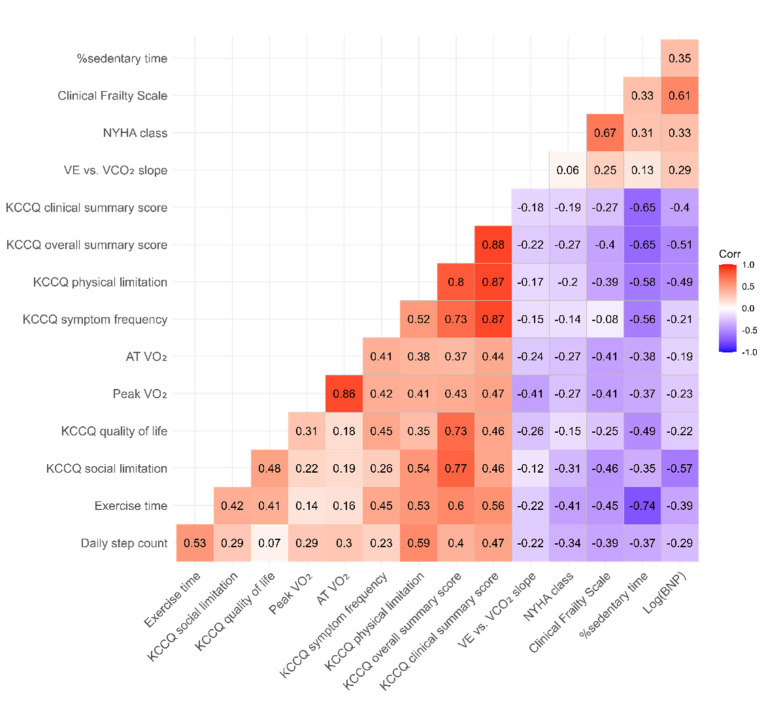
Correlation heatmap of the Kansas City Cardiomyopathy Questionnaire scores, accelerometer-measured parameters, cardiopulmonary exercise testing, BNP, NYHA class, and Clinical Frailty Scale. Values in cells show Pearson correlation coefficients. KCCQ = Kansas City Cardiomyopathy Questionnaire; VO_2_ = oxygen consumption; VCO_2_ = carbon dioxide production; VE = expiratory minute volume; AT = anaerobic threshold; NYHA = New York Heart Association. Variables were ordered based on hierarchical clustering using hclust in R.

**Figure 3 jcm-10-04103-f003:**
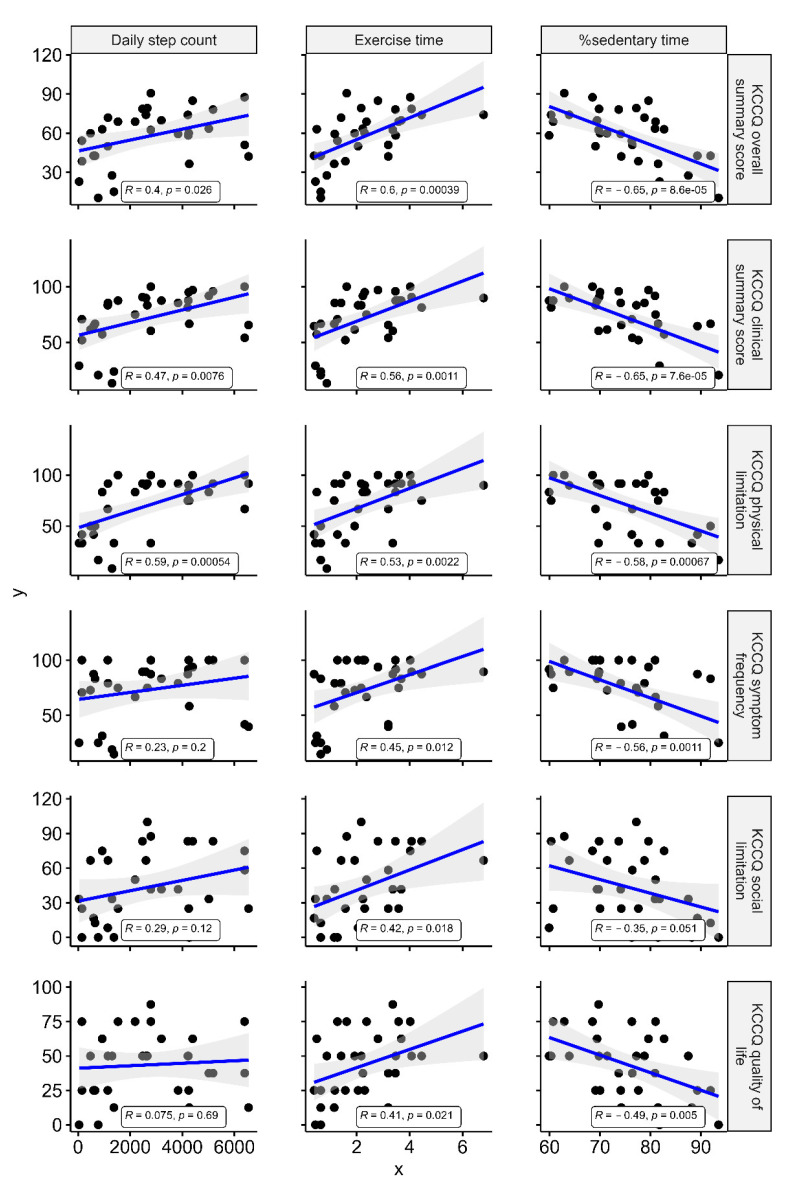
Correlation of accelerometer-measured physical activity with the Kansas City Cardiomyopathy Questionnaire. R represents Pearson’s correlation coefficient. KCCQ = Kansas City Cardiomyopathy Questionnaire; OSS = overall summary score; CSS = clinical summary score.

**Figure 4 jcm-10-04103-f004:**
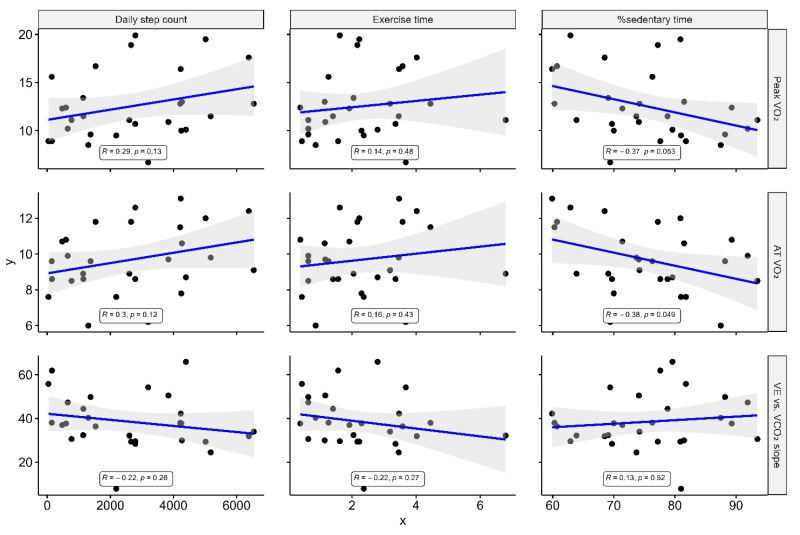
Correlation of accelerometer-measured physical activity with CPX parameters. R represents Pearson’s correlation coefficient. VO_2_ = oxygen consumption; VCO_2_ = carbon dioxide production; VE = expiratory minute volume; AT = anaerobic threshold.

**Figure 5 jcm-10-04103-f005:**
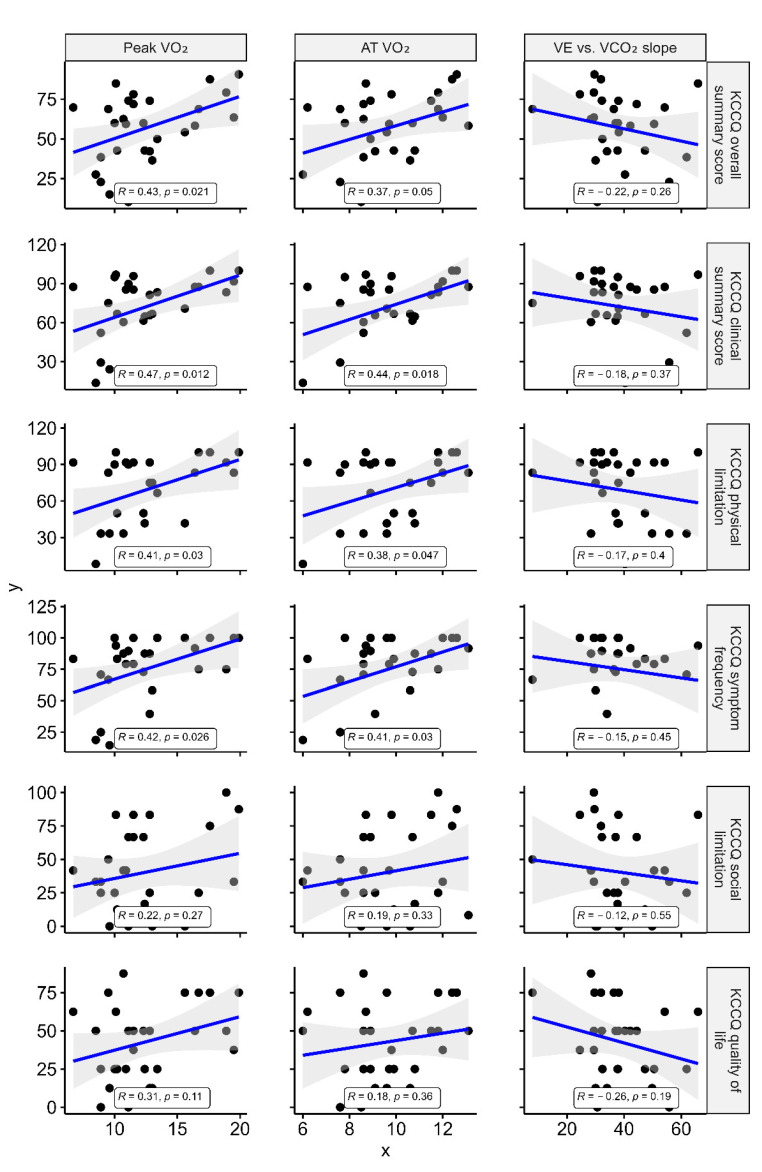
Correlation of CPX parameters with the Kansas City Cardiomyopathy Questionnaire. Caption: R represents Pearson’s correlation coefficient. KCCQ = Kansas City Cardiomyopathy Questionnaire; OSS = overall summary score; CSS = clinical summary score; VO_2_ = oxygen consumption; VCO_2_ = carbon dioxide production; VE = expiratory minute volume; AT = anaerobic threshold.

**Table 1 jcm-10-04103-t001:** Baseline characteristics.

Variable	All Patients *n* = 31	KCCQ-OSS ≥ 60 *n* = 16	KCCQ-OSS < 60 *n* = 15	*p*-Value
Background				
Age, years	63 (51–78)	63 (44–69)	72 (56–78)	0.151
Men, *n* (%)	20 (65)	11 (69)	9 (60)	0.611
Body mass index, kg/m^2^	20.8 (18.9–24.9)	24.2 (21.2–27.8)	20.3 (18.5–20.7)	0.001
Systolic blood pressure, mm Hg	97 (90–105)	98 (85–105)	97 (91–112)	0.599
Heart rate, beat per minute	70 (60–75)	70 (60–76)	71 (60–75)	0.770
LVEF, %	32 (24–42)	27 (24–37)	37 (29–58)	0.041
LVDd, mm	58 (52–66)	58 (54–67)	56 (42–61)	0.161
LVDs, mm	50 (41–58)	52 (44–64)	46 (29–54)	0.045
NYHA functional class, *n* (%)				0.354
II	8 (26)	3 (19)	5 (33)	
III	23 (74)	13 (81)	10 (67)	
Clinical frailty scale, *n* (%)				0.038
2 (fit)	12 (39)	10 (62)	2 (13)	
3 (managing well)	11 (36)	4 (25)	7 (47)	
4 (very mild frailty)	7 (23)	2 (12)	5 (33)	
5 (mild frailty)	1 (1)	0 (0)	1 (7)	
Triggers of acute decompensation				0.610
Nonadherence to diet, *n* (%)	4 (13)	2 (13)	2 (13)	
Nonadherence to medication, *n* (%)	1 (3)	1 (6)	0 (0)	
Overwork, *n* (%)	4 (13)	1 (19)	3 (20)	
Arrhythmia, *n* (%)	5 (16)	3 (19)	2 (13)	
Coronary ischemia, *n* (%)	2 (7)	1 (6)	1 (6)	
Infection, *n* (%)	3 (10)	2 (13)	1 (6)	
Others or none, *n* (%)	15 (48)	8 (57)	7 (47)	
Comorbidities				
≥1 HF hospitalization in the past year, *n* (%)	21 (68)	12 (75)	9 (60)	0.372
Coronary artery disease, *n* (%)	6 (19)	2 (13)	4 (25)	0.411
Atrial fibrillation, *n* (%)	10 (32)	5 (31)	5 (33)	0.901
Hypertension, *n* (%)	12 (39)	9 (56)	3 (20)	0.038
Diabetes mellitus, *n* (%)	8 (26)	3 (19)	5 (33)	0.354
Stroke, *n* (%)	7 (23)	4 (25)	3 (20)	0.739
COPD, *n* (%)	2 (7)	1 (6)	1 (7)	0.962
Laboratory tests				
Hemoglobin, g/dL	12.8 (11.3–14.2)	13.0 (11.4–14.9)	12.4 (11.2–13.8)	0.572
Creatinine, mg/L	1.1 (0.9–1.5)	1.1 (0.9–1.4)	1.2 (0.9–1.5)	0.520
Blood urea nitrogen, mg/L	24.6 (19.2–34.4)	20.6 (14.8–30.5)	30.0 (24.6–35.5)	0.027
Sodium, mEq/L	138.9 (136.6–140.6)	138.7 (136.7–140.0)	138.9 (136.1–140.9)	0.545
Potassium, mEq/L	4.3 (4·0–4·6)	4.3 (4.2–4.5)	4.3 (3.9–4.6)	0.545
Total bilirubin, mg/L	0.9 (0.7–1.3)	0.9 (0.7–1.2)	0.9 (0.6–1.3)	0.861
Albumin, mg/L	3.9 (3.6–4.1)	4.0 (3.8–4.1)	3.8 (3.5–4.1)	0.318
BNP at hospitalization, pg/mL	1118 (447–1498)	835 (248–1173)	1478 (573–1958)	0.022
BNP at discharge, pg/mL	370 (244–437)	253 (160–397)	413 (334–788)	0.001
BNP improvement, %	58.4 (28.4–74.9)	65.1 (26.8–78.8)	56.0 (20.9–71.8)	0.458
HF treatment				
Loop diuretics, *n* (%)	27 (88)	12 (75)	15 (100)	0.038
ACEI or ARB, *n* (%)	23 (74)	12 (75)	11 (73)	0.916
ARNI, *n* (%)	0 (0)	0 (0)	0 (0)	–
Beta blocker, *n* (%)	27 (88)	14 (88)	13 (87)	0.945
MRA, *n* (%)	20 (65)	11 (69)	9 (60)	0.611
SGLT2i, *n* (%)	11 (35)	5 (31)	6 (40)	0.611
ICD, *n* (%)	3 (10)	2 (13)	1 (7)	0.583
CRT, *n* (%)	3 (10)	2 (13)	1 (7)	0.583
CPX parameters *				
Peak VO_2_, mL/kg/min	11.5 (10.0–15.0)	11.5 (10.1–17.9)	11.7 (9.4–13.1)	0.427
AT VO_2_, mL/kg/min	9.6 (8.6–11.3)	9.4 (8.4–11.9)	9.6 (8.6–10.6)	0.701
VE vs. VCO_2_ slope	37.4 (30.2–46.6)	32.0 (29.2–39.6)	39.2 (33.6–50.0)	0.050
Accelerometer-measured parameters				
Daily step count, *n*/day	2604 (926–4247)	2792 (2259–4360)	1138 (475–4230)	0.049
Exercise time	2.2 (1.2–3.5)	3.1 (2.2–3.9)	1.2 (0.6–2.0)	0.001
%sedentary time, %	76.3 (69.4–81.5)	69.9 (65.1–79.4)	77.6 (74.2–88.2)	0.030

Values are median (interquartile range). All variables were measured at the time of discharge other than BNP at hospitalization. BNP improvement was defined as (BNP at hospitalization—BNP at discharge)/BNP at hospitalization; Notes: * CPX was performed in 28 patients. Abbreviations: LVEF, left ventricular ejection fraction; LVDd, left ventricular end-diastolic diameter; LVDs, left ventricular end-systolic diameter; NYHA, New York Heart Association; HF, heart failure; COPD, chronic obstructive pulmonary disease; BNP, B-type natriuretic peptide; ACEI, angiotensin-converting enzyme inhibitor; ARB, angiotensin receptor antagonist; ARNI, angiotensin receptor neprilysin inhibitor; MRA, mineralocorticoid receptor antagonist; SGLT2i, sodium glucose cotransporter-2 inhibitor; ICD, implantable cardioverter defibrillator; CRT, cardiac resynchronization therapy; CPX, cardiopulmonary exercise testing; VO_2_, oxygen consumption; AT, anaerobic threshold; VE, expiratory minute volume; VCO_2_, carbon dioxide production.

## Data Availability

The data presented in this study are available on reasonable request from the corresponding author.

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
