# Peer review of "Assessment of Physical Activity Using Waist-Worn Accelerometers in Hospitalized Heart Failure Patients and Its Relationship with Kansas City Cardiomyopathy Questionnaire"

_jcm, 2021, doi:10.3390/jcm10184103_

Round 1

Reviewer 1 Report

This manuscript examines the relationship between measured physical activity and subjective well-being in heart failure patients. The study has shortcomings in content and form:

1) The methodology should include a summary chart of when what is measured.
2) Data on the outcome (re-hospitalization? death? decompressions?) are missing.
3) Important information about the cohort is missing: LVEDD? Volumes? etiology of decompensation?
4) Is the inpatient stay really the right time period to determine the impact of an accelerometer? Is this really representative of the patient?

Reviewer 2 Report

In their paper, the Authors focused on a very actual and significant topic such as the assessment of physical activity in heart failure (HF) patients.

This is a very argued issue and it has multiple interesting potential applications in real-life HF patients management.

I believe that the following points should be addressed before consideration for acceptance and publication.

  1. Introduction

The Authors should better and briefly define the benefits of physical activity and exercise on HF; moreover, they should mention the current international recommendations about physical activity in patients with HF. Finally, it is not clear if the study was specifically focused on patients with HF with reduced ejection fraction or not.

  1. Materials and methods
    • Study cohort and sample: Authors should specify the subset of patients with HF they focused on, according to international HF guidelines classification. Authors should here mention the subgroup of patients who underwent right heart catheterization (RHC); moreover, they should add a specific section about the materials and methods related to RHC. Finally, inclusion and exclusion criteria should be reported.
    • Clinical data acquisition: did the Authors consider any comorbidities potentially limiting physical activity?
    • Accelerometry: What were the wear time criteria to consider valid registrations? How did the Authors define sedentary time? The accelerometer-derived parameters should be accurately delineated, as well as: criteria for defining non-wear time, minimum number of minutes needed to be considered a valid day, and number of valid days needed for a patient to be included in the analysis.

Which are the patient-reported outcomes that the Authors mention in the title? Authors should discuss them (or I suggest modifying the title more clearly).

  1. Discussion

How did the Authors identify the patients “who were sufficiently decongested”?

The Authors should discuss the controversies regarding 1) the use of accelerometers in the assessment of physical activity and sedentary behaviours in HF patients (Vetrovsky T.et al, Advances in accelerometry for cardiovascular patients: a systematic review with practical recommendations, ESC Heart Failure 2020; 7: 2021–2031, DOI: 10.1002/ehf2.12781); 2) the application of generic physical activity thresholds in HF patients (Dibben, G.O., Gandhi, M.M., Taylor, R.S. et al. Physical activity assessment by accelerometry in people with heart failure. BMC Sports Sci Med Rehabil 12, 47 (2020). https://doi.org/10.1186/s13102-020-00196-7).

Round 2

Reviewer 1 Report

Although the work does not provide any major new insights, it is well executed. The data are well presented and comprehensible.

Reviewer 2 Report

I was pleased to have the opportunity to review this manuscript. 

I believe that the Authors did a good review job.